# Transfer Learning with Causal Counterfactual Reasoning in Decision Transformers

## Abstract

The ability to adapt to changes in environmental contingencies is an important challenge in reinforcement learning. Indeed, transferring previously acquired knowledge to environments with unseen structural properties can greatly enhance the flexibility and efficiency by which novel optimal policies may be constructed. In this work, we study the problem of transfer learning under changes in the environment dynamics. In this study, we apply causal reasoning in the offline reinforcement learning setting to transfer a learned policy to new environments. Specifically, we use the Decision Transformer (DT) architecture to distill a new policy on the new environment. The DT is trained on data collected by performing policy rollouts on factual and counterfactual simulations from the source environment. We show that this mechanism can bootstrap a successful policy on the target environment while retaining most of the reward.

## 1 Introduction

Reinforcement learning (RL) is a powerful sequential decision-making framework (Sutton & Barto, 2018). A pervasive challenge is the difficulty of adapting trained agents between different scenarios. For instance, consider robots operating a production line. RL can be used to train a robot to perform a certain task on a single line. If the conditions in this production line change, e.g., change of light sensors on the robot (observation/state), change of product specs (reward), the previously acquired policy might not be applicable anymore and the robot needs to be retrained in the new environment. Transfer learning can help in this scenario whereby the trained agent's knowledge can be incorporated into a new agent operating under perturbations to the environment dynamics. This can significantly reduce the retraining costs of the new agent and in some instances obviate the need for re-training altogether.

In this work, we explore transfer in reinforcement learning in the offline setting (Levine et al., 2020) using causal reasoning (Pearl, 2009). Offline RL, whereby sampled environment state and action trajectories are used to train an agent in the absence of adaptive exploration, has become a dominant data-driven paradigm for large-scale real-world RL applications. We specifically focus on the problem of training agents such that they are robust to structural changes in the environment. We leverage the causal knowledge of a source environment's structure to generate a set of counterfactual environments that the original agent could potentially have encountered in order to collect data to train a new more general offline learning agent. We hypothesize that imbuing agents with knowledge of possible counterfactual environments may aid in regularizing the agent's internal representation of environment contingencies thereby making them more adaptive to structural changes.

## 2 Background and Definitions

We assume that the reader is familiar with the basic concepts of Reinforcement Learning (RL) (Sutton & Barto, 2018). An *environment* $\mathcal{E}$ is defined as a Markov decision process (MDP) with components $\{S, A, p, r, \gamma\}$, where $S$ is the set of states $s \in S$, $A$ is the set of actions $a \in A$, $p$ is the transition function, $r$ is the reward function, and $\gamma$ is the discount factor. An agent seeks to learn a policy $\pi : S \to A$ that maximises the total discounted reward.

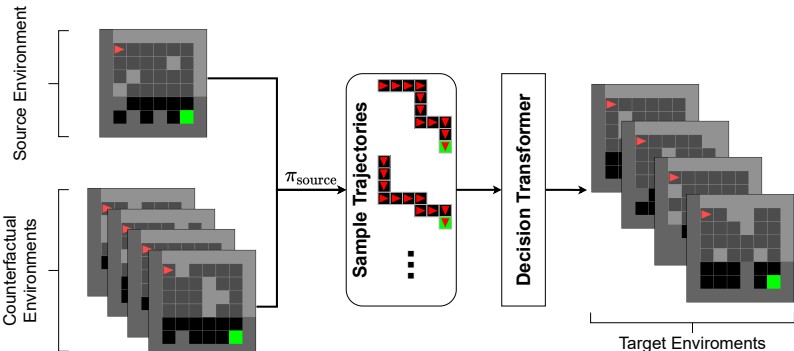

Figure 1: Illustration of of the proposed training scheme on the `gym_minigrid` environment suite. Sample trajectories are generated from $\pi_{\text{SOURCE}}(a_t|s_t)$ by performing rollouts on the original and the counterfactual environments. The trajectories are used to train a DT that generalises well on a possibly harder set of target environments.

*Offline reinforcement learning* is a data-driven approach to the standard RL problem, where an optimal policy is learned from a static offline dataset of transitions, $\mathcal{D} = \{s_t^i, a_t^i, s_{t+1}^i, r_t^i\}_{t=1...T, i=1...N}$ where $T$ is the rollout horizon and $N$ is the number of sample trajectories. The objective of offline RL is identical to the standard RL objective, i.e. maximising the sum of future discounted rewards. However, its learning mechanism is different as the policy is not allowed to interact with the environment during training (Levine et al., 2020).

In this work, we implement offline RL using the transformer deep neural network architecture which has been introduced to RL as the *Decision Transformer (DT)* (Chen et al., 2021; Janner et al., 2021). The DT is a sequence-to-sequence model that outputs action sequences via inference over trajectories. The model consumes offline trajectories comprised of a $K$-long sequence of returns-to-go, states/observation and actions triplets, $\tau = (R_t, s_t, a_t, \ldots, R_{t+K}, s_{t+K}, a_{t+K})$, where $R_t = \sum_{i=t}^{T} r_i$ and $r_t$ is the reward at time $t$. These trajectories can be obtained from a demonstrator or by performing multiple rollouts of a trained policy on some environment.

We assume that we are given a *source policy* $\pi_{\text{SOURCE}}(a_t|s_t)$ that has been trained on a source environment $\mathcal{E}_{\text{SOURCE}}$. The idea of *transfer learning* in RL is to use the information that was learned in $\pi_{\text{SOURCE}}(a_t|s_t)$ to derive a new policy $\pi_{\text{TARGET}}(a_t|s_t)$ that performs well on a new target environment $\mathcal{E}_{\text{TARGET}}$. In this work, we focus on the specific type of transfer learning, specifically, the problem of transferring policies between environments with different *dynamics*. An environment is specified by a transition function and a reward function, i.e., $\mathcal{E} = (p(s'|s,a), r(s,a))$. We assume that the reward function is kept constant while the transition function varies between the source and the target environments, i.e. $p_{\text{SOURCE}}(s'|s,a) \neq p_{\text{TARGET}}(s'|s,a)$ for $\mathcal{E}_{\text{SOURCE}} = (p_{\text{SOURCE}}(s'|s,a), r(s,a))$ and $\mathcal{E}_{\text{TARGET}} = (p_{\text{TARGET}}(s'|s,a), r(s,a))$. This problem is particularly challenging in the model-free setting, since we do not learn an explicit representation of the transition structure of the environment and, instead, we rely on eliciting an implicit representation internally within the DT.

*Causal counterfactual reasoning* is going back to Hume (1739) and has been a subject of active research in the last years. Essentially, an event $A$ is a cause of an event $B$ if $A$ happened before $B$ and in a closest possible world where $A$ did not happen, $B$ did not happen either. In this work, we adapt these concepts to RL, where an event $A$ is a decision of a given RL policy in a given state and $B$ is the success in achieving the reward.

## 3 METHODOLOGY

We address the problem of transfer learning under a change of dynamics by training a general offline DT agent using the source policy as a demonstrator. We obtain demonstration trajectories for the DT by performing multiple rollouts of the source policy on the source environment. However, these trajectories are unlikely to contain enough variability to generalise on the target environments.

Therefore, we enrich the demonstration dataset by rolling out the source agent on a set of counterfactual environments. Producing simulations from the counterfactual environment can aid in improving the current policy by indirectly incorporating information about alternative environment structures. This combined dataset is then used to train a DT to synthesize a new successful policy that performs well on the unseen target environments. An illustration of this idea is shown in Figure 1.

### 3.1 COUNTERFACTUAL ENVIRONMENTS AS STRUCTURAL INTERVENTIONS ON THE SOURCE ENVIRONMENT

In many cases, the change in the transition function between the source and target environments can be induced by a structural change in the source environment features, e.g., erecting walls in a minigrid environment or changing the physics engine in a robotics simulation. In such cases, one can model these types of changes as *interventions*, in the causal sense (Pearl, 2009), on the environment features that induce a new transition function. Mathematically, let $\Phi$ denote some structural (and stationary) features of the environment; the transition function $p(s'|s, a, \Phi)$ therefore describes a family of environments. By intervening on $\Phi$, on can arrive at a specific member of this family, e.g., $p_{\text{SOURCE}}(s'|s, a) = p(s'|s, a, \text{do}(\Phi = \phi_{\text{SOURCE}}))$.

One can generate a set of counterfactual environments $\{\mathcal{E}_{\text{CF}}^i\}_{i=1}^N$ by performing the interventions $\text{do}(\Phi = \phi_{\text{CF}}^i)$ for $\phi_{\text{CF}}^i \sim p(\phi_{\text{CF}})$.

To obtain counterfactual trajectories, we generate a set of counterfactual environments by performing interventions on the structural level, and then perform rollouts of the source policy on the set of counterfactual environments.

### 3.2 WEIGHTING SUCCESSFUL TRAJECTORIES BY THE AVERAGE TREATMENT EFFECT

Modelling counterfactual environments as interventions allows us to use the notion of a *treatment effect* on the trajectory distribution to describe the impact of these interventions on the source policy measured by the difference in total accumulated reward between the source (factual) and the counterfactual environments (Pearl, 2009):

$$\text{ATE}[\phi_{\text{CF}}] = \sum_{t=1}^{T} \mathbb{E}_{p_{\mathcal{E}_{\text{CF}}}(\tau)}[r_t] - \mathbb{E}_{p_{\mathcal{E}_{\text{SOURCE}}}(\tau)}[r_t] \;, \tag{1}$$

where

$$\begin{aligned} p_{\mathcal{E}_{\text{CF}}}(\tau) =& p(s_1)p(r_1|s_1)\pi_{\text{SOURCE}}(a_1|s_1)p(s_2|a_1, s_1, \text{do}(\Phi = \phi_{\text{CF}}))p(r_2|s_2)\dots \\ &\pi_{\text{SOURCE}}(a_{T-1}|s_{T-1})p(s_T|a_{T-1}s_{T-1}, \text{do}(\Phi = \phi_{\text{CF}}))p(r_T|s_T), \end{aligned} \tag{2}$$

and similarly for $p_{\mathcal{E}_{\text{SOURCE}}}(\tau)$.

A popular and demonstrably productive heuristic in offline RL is to weight sample trajectories during training depending on their likely contribution to the training objective. For example, this has previously been referred to as prioritized replay in DQN agents (Schaul et al., 2015). Furthermore, such mechanisms appear to be implemented in biological agents in the form of neural replay prioritization (Mattar & Daw, 2018) and counterfactual reasoning aimed at detecting environment shifts (Zhang et al., 2015). In the present work, we used the ATE measurement (Eqn. 1) to rank the counterfactual environments according to their effect on the source policy. Correspondingly, in our training procedure, we can use ATE to bias the trajectory sampling whereby more successful trajectories are over-weighted when training the DT.

## 4 EXPERIMENTS

We tested our method on the `gym_minigrid` environment suite (Chevalier-Boisvert et al., 2018). We created a new environment `RandomObstaclesMinigrid`[1], which randomly generates a configuration of a fixed number of obstacles (walls) inside an `EmptyGrid`. Samples from this environment can be seen in the illustration in Figure 1.

---

[1] The code for this environment as well as the experiment code are publicly available at `anonymisedURL`

Table 1: Experimental Setup

| Scenario | Easy | Hard |
|---|---|---|
| Source Environments | 1 obstacle configuration [6 obstacles] | 1 obstacle configuration [6 obstacles] |
| Counterfactual Environments | 2000 obstacle configurations [6 obstacles] | 2000 obstacle configurations [6 obstacles] |
| Target Environments | 1000 obstacle configurations [6 obstacles] | 1000 obstacle configuration [7 obstacles] |

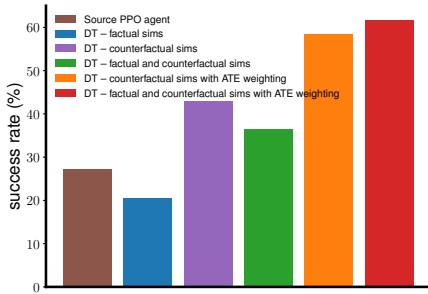 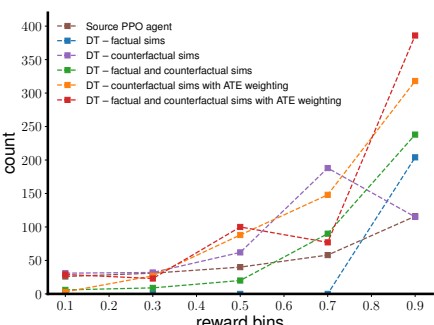

Figure 2: (a) The rate of goal attainment for different agents on the `RandomObstaclesMinigrid` with six obstacles. (b) Comparison of different transfer mechanisms on a set of target environments with six obstacles. The original PPO policy and the DT policy are generally less capable at solving the problem and achieving a positive reward than the DT agents trained with counterfactual simulations. Weighting the counterfactual DT agents with the ATE weighting scheme in Section 3.2 improves the generalisation ability of those agents.

Our experimental objective is to transfer a PPO policy that is trained on a single configuration of obstacles, representing the *source* environment, to a set of unseen randomly generated obstacle configurations, representing a set of *target* environments. We test this in two settings representing different levels of difficulty for the transfer learning problem. The first is an easy setting, where the source and the target environments have the same number of obstacles. The second is the harder setting, where the target environments contain a larger number of obstacles than the source environment.

We compare different solutions to the transfer learning problem:

a) Using the original PPO policy directly on the target environments.
b) Using a DT policy trained on factual simulations from the source environment.
c) Using a DT policy trained on counterfactual simulations only.
d) Using a DT policy trained on the both factual and counterfactual simulations.

Furthermore, when using the counterfactual simulations, we experiment with using the ATE weighting scheme described in Section 3.2.

The counterfactual simulations are generated by performing multiple rollouts of the PPO source policy on the `RandomObstaclesMinigrid`, varying the obstacles configuration in every episode. The counterfactual environments are seeded differently than the target environment to avoid data leakage from training to testing. Furthermore, for the hard scenario the counterfactual simulations are set to have the same number of obstacles as the source environment, i.e., less obstacles than the target environments. Table 1 summarises our experimental setup.

To ensure variability in the counterfactual trajectories, we implement a fail-safe feature in our simulations, whereby if the PPO agent is stuck (i.e. bumps into a wall), it reverts to exploration for a certain number of steps (set heuristically to 10).

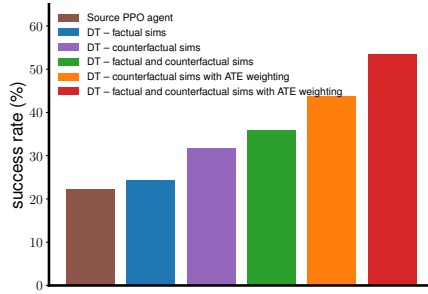 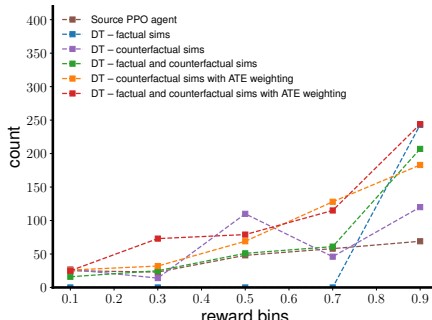

Figure 3: (a) The rate of goal attainment for different agents on the `RandomObstaclesMinigrid` with seven obstacles. (b) Comparison of different transfer mechanisms on a set of target environments with seven obstacles. Similar to the six obstacles scenario, the DT policies with factual and counterfactual simulations and ATE weighting outperform those with other transfer mechanisms.

## 5 RESULTS

The results of our experiments are summarised in Figure 2 and Figure 3 (and in Table 2 & Table 3 in Appendix A). In both scenarios, the DT agents that are trained on both factual and counterfactual simulations outperform the other agents. The ATE weighting scheme in Section 3.2 is shown to be effective.

As the distributions of rewards acquired are bimodal (see Appendix A for the details), we examine them further in Figure 2 and Figure 3. For the easy scenario presented in Figure 2, the left plot a) shows the percentage of episodes in which the agent reaches the goal. The original PPO policy and the DT policy trained on factual simulations only are only able to succeed on less than $30\%$ of the target environments. Incorporating counterfactual simulations in the DT training improves the overall performance of the DT on the target environments. The addition, the ATE weighting scheme enhances the DT agent further by over-emphasising successful trajectories. Looking at the histogram of the positive rewards in the right plot b), we can see the success of the proposed training scheme, where the weighted DT agent trained on factual and counterfactual simulations achieves the highest possible rewards for the majority of the evaluation episodes.

On the hard scenario, similar observations can be seen in Figure 3, where the proposed training scheme achieves the highest performance.

## 6 CONCLUSIONS AND FUTURE WORK

We presented a novel training scheme for DTs that leverages counterfactual reasoning to transfer information from a source agent to a new offline agent that generalises to unseen target environments. We conceptualized this as performing structural interventions on the source environment features to obtain a set of counterfactual environments. This is generally possible in simulated systems. Furthermore, we showed that weighting the trajectories by the ATE on the total reward, improves the generalisation ability of the DT as it learns to reconstruct more successful trajectories. We empirically demonstrated the efficacy of our proposal on the `gym_minigrid` environment suite.

In this work, we adopted a simple specification of the counterfactual trajectories. They were generated by performing rollouts of the source policy on alternative environments drawn from the generative model of the source environment. Thus, our method relies on having access to a veridical environment simulator (or more generally the structural equation model of the environment). This assumption is, of course, unrealistic for most problems. In the future, we will investigate solutions for generating more realistic counterfactual trajectories that can be used to enrich offline learning agents. Our current work can be thought of as a proof of concept showing the efficacy of the special case where full knowledge of the generative structure of the environment is available.

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

# A  DETAILED RESULTS

Table 2: Results summary on the `RandomObstaclesMinigrid` target with six obstacles.

| Agent | Average reward | Average episode length |
|---|---|---|
| Source PPO agent | -0.55 | 211.56 |
| DT – factual sims | -0.60 | 206.60 |
| DT – counterfactual sims | -0.29 | 185.88 |
| DT – factual and counterfactual sims | -0.34 | 180.14 |
| DT – counterfactual sims with ATE weighting | 0.03 | 142.34 |
| DT – factual and counterfactual sims with ATE weighting | 0.09 | 134.60 |

Table 3: Results summary on the `RandomObstaclesMinigrid` target with seven obstacles.

| Agent | Average reward | Average episode length |
|---|---|---|
| Source PPO agent | -0.64 | 169.84 |
| DT – factual sims | -0.53 | 151.56 |
| DT – counterfactual sims | -0.47 | 156.02 |
| DT – factual and counterfactual sims | -0.36 | 143.04 |
| DT – counterfactual sims with ATE weighting | -0.26 | 136.87 |
| DT – factual and counterfactual sims with ATE weighting | -0.09 | 124.24 |

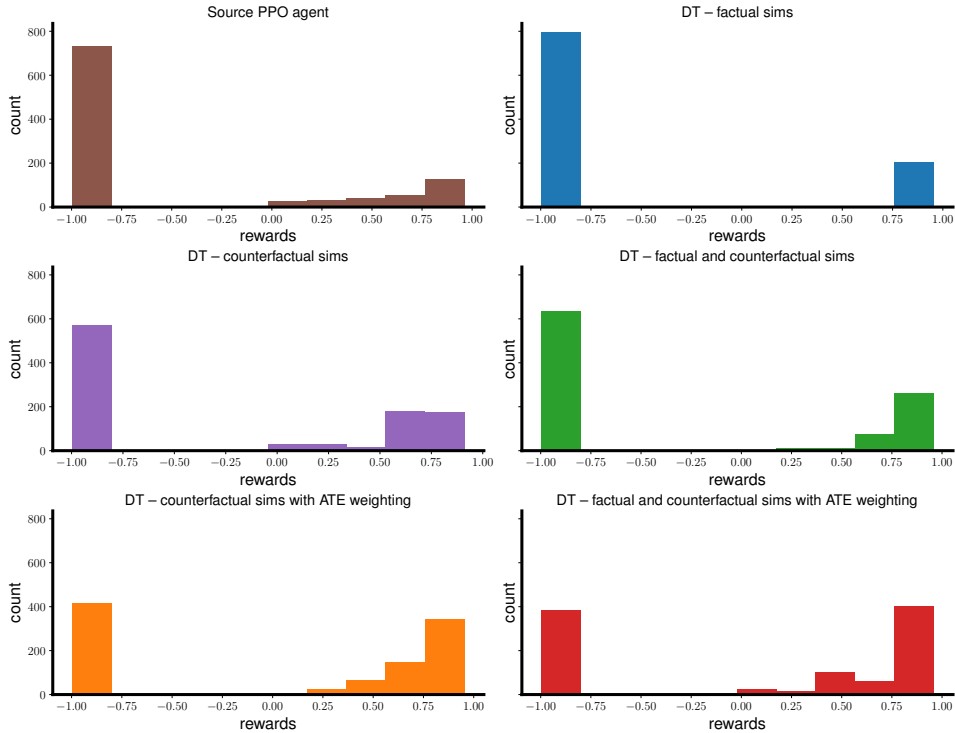

Figure 4: Detailed distribution of the rewards attained on the target environments for the easy scenario (6 obstacles). There are 1000 obstacle configurations in total.

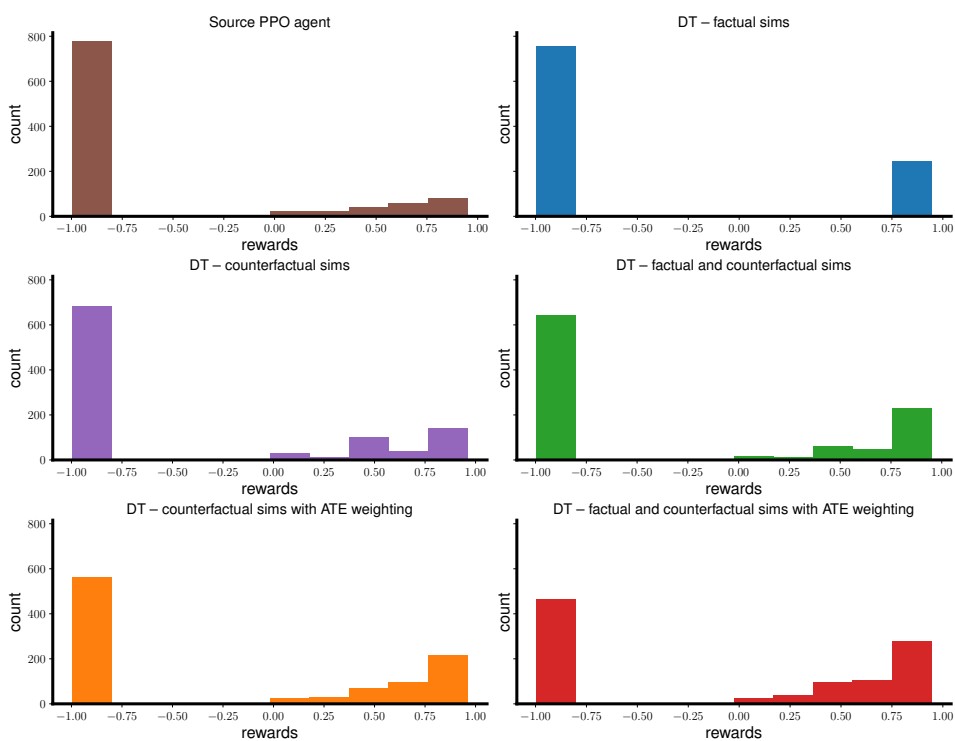

Figure 5: Detailed distribution of the rewards attained on the target environments for the hard scenario (7 obstacles). There are 1000 obstacle configurations in total.

