# OpenReview forum: "Transfer Learning with Causal Counterfactual Reasoning in Decision Transformers"
_ICLR.cc/2022/Workshop/OSC — Submitted to ICLR2022 OSC _

### Official Review · Reviewer_hS6T · 2022-03-15
**Good application of ATE and counterfactual simulations for transfer/adaptation/generalization for offline RL**

**Rating:** 2
**Confidence:** 3

**Review:**

This work studies the problem of transfer learning/adaptation/generalization in offline RL algorithms. The main idea is to train the offline RL policy (instantiated as a Decision Transformer) on a more diverse set of policy rollouts generated from counterfactual environments. Further, they also use the concept of average treatment effect (ATE), here defined as the difference in cumulative returns of a policy rollout on the source environment and counterfactual (CF) one as a weighting factor used to proportionately sample trajectories (akin to Prioritized Experience Replay) for learning policy updates.

Pros:

1) The main ideas are simple and well-motivated. The writing and elucidation of the main ideas is clear and concise.
2) The research question being investigated is topical and very relevant to the workshop.
3) Ablation experiments showing the effect of both components i.e. counterfactual trajectories and non-uniform sampling based on ATE to allow the policy to generalize on unseen environments.

Cons:

1) Experimental results only shown on 1 newly created "RandomObstacles" environment from the Minigrid suite.

2) Crucially, this method assumes that the underlying generative process of an environment is accessible. How can this dependence be overcome?

3) Source and target environments in this work are only assumed to differ in the transition function but have the same reward function, which partially allows us to continue to use the source policy to generate CF rollouts from the CF environments. Despite this the authors still use an exploration “fail-safe” strategy when the source policy is stuck. How can the counterfactual data collection process be extended to a more natural setting where the transition model remains the same but the reward function differs across source and target environments? For example, in the physical world, Newtonian mechanics remains consistent but the tasks we would like to achieve varies.

---

### Official Review · Reviewer_rSUD · 2022-03-15
**Empirical contribution is simplified too much to be of relevance to the workshop**

**Rating:** 1
**Confidence:** 3

**Review:**

### Summary

This paper focuses on training offline RL agents such that they are robust to structural changes in the environment. In leveraging causal knowledge of a source environment's structure, a set of counterfactual environments is obtained to generate additional data for the offline agent. Here the offline agent is implemented as a Decision Transformer (DT), and the target environments to which we wish to generalize are assumed to differ only in their transition dynamics to the source environment (reward functions are the same).

Given access to a source policy trained on the source environment, it is proposed to collect data under this policy on both the source and a set of counterfactual environments to enrich the offline data available to the DT. This lets it synthesize a new policy that performs well on unseen target environments. Counterfactual environments are obtained through intervention on structural environment features (eg. erect walls). In this way, the notion of "treatment effect" on the trajectory distribution can be used to measure the impact of interventions on the source policy as the difference in total accumulated reward between source (factual) and counterfactual environments. This metric, abbreviated as ATE, is then used as a heuristic to bias the trajectories the DT receives during (over-weight more successful trajectories with large ATE).

The experiments focus on transferring a PPO policy trained on a source environment based on the EmptyGrid MiniGrid environment but having a fixed configuration of obstacles in it. Two settings are considered for the target environments having randomly generated obstacles: (1) where the number of obstacles is the same, and (2) where the target environments include additional obstacles. Counterfactual environments also contain randomly generated obstacles but using a different seed and the same number of obstacles to the source environment in the harder setting. It can be observed that the additional data from the counterfactual environments as well as the ATE weighting leads to improved performance on the source environment and in the transfer setting. On the harder setting (using 7 in stead of 6 obstacles during transfer) similar trends can be seen.

### Review

While the motivation of the counterfactual training scheme explored can be related to the workshop theme, the actual setting considered is simplified to the extent that it is no longer relevant to the workshop theme. To be more specific, in essence this paper explores using two data sources (one additional compared to normal) for training a Decision Transformer:

(1) data collected by executing a source policy in an empty MiniGrid environment having a particular configuration of 6 randomly configured walls. The source policy was trained on this environment as well.
(2) data collected by executing the same source policy on identical environments that differ only in the random configuration of the 6 walls

The main findings are that including (2) as a type of data augmentation improves performance on target environments that follow the same distribution of 6 randomly sampled wall (easy case) or that include one additional randomly placed wall (hard case). While it is quite clear that using (1) and (2) as data sources should lead to improved performance, these settings are too simplistic to say anything about whether counter-factual data augmentation in this way should lead to improvement in the general case. In particular, different types of interventions are not explored, and neither a setting where counter-factual, source, and target  environments are much more different.
Even for the current environments, I'd encourage the authors to conduct a wider search on the number of random walls they contain and explore different backbones from MiniGrid other than EmptyGrid (eg. environments with key corridors, etc.).

Regarding the current results, one thing that stood out in Figure 2 is that adding factual data in fact may not always lead to clear improvements (eg. purple vs. green bar or table 2), while in general most improvement appears to be driven by the heuristic ATE weighting. I did not find that particularly intuitive and it would be good to clarify/analyze this. More generally, no error bars are reported for these results, which unfortunately makes it difficult to draw meaningful conclusions.

---

### Decision · Program_Chairs · 2022-03-21

**Decision:**

Reject

**Comment:**

Unfortunately, the reviewers found the paper to be not ready for presentation at the workshop. I recommend the authors to take a closer look at the review from rSUD and address the experimental issues they raise. In particular, it should be clear whether the improvement comes from the counterfactual data.